# Knowledge and Attitude of First-Aid Treatments for Snakebites, and the Perception of Snakes among the Students of Health Sciences at Jazan University, Saudi Arabia

**DOI:** 10.3390/healthcare10112226

**Published:** 2022-11-07

**Authors:** Saad S. Alqahtani, David Banji, Otilia J. F. Banji, Mamoon H. Syed, Nabeel Kashan Syed, Abdulkarim M. Meraya, Ahmed A. Albarraq, Hilal A. Thaibah, Amani Khardali, Ibrahim A. Alhomood, Aeshah M. Mahzari, Omaymah M. Alshlali

**Affiliations:** 1Department of Pharmacy Practice, College of Pharmacy, Jazan University, Jazan 45142, Saudi Arabia; 2Pharmacy Practice Research Unit, College of Pharmacy, Jazan University, Jazan 45142, Saudi Arabia; 3Department of Pharmacology & Toxicology, College of Pharmacy, Jazan University, Jazan 45142, Saudi Arabia

**Keywords:** snakebite, first aid, students, venomous, perception, Saudi Arabia

## Abstract

First aid is the most basic and life-saving service provided before emergency care is received. This study aimed to assess students’ knowledge and attitudes about performing first aid for snakebite emergencies and their perception of snakes. A cross-sectional study was conducted between January and March 2019 among upper-level students (3rd year and above) of health-related courses at Jazan University, Saudi Arabia. Three hundred and nine students from four programs responded to the self-administered questionnaire. The collected data were analyzed using SPSS 23.0. The highest percentage of the study sample comprised pharmacy students (44%), followed by nursing (25.2%), medical (16.8%), and emergency medicine students (13.9%). Overall, the emergency medicine students exhibited greater knowledge of snakebite first aid. However, over three-fourths of the students were unaware of dry bites. Around two-thirds were sure that they should not massage the bite site, while nearly one-fourth were not sure about the use of a tourniquet. The fifth- and sixth-year students had extremely diverse perspectives on snakes. The majority of the participants (92.8%) did not feel good about snakes. However, most of the students (95.2%) wanted to learn about snakebite first aid and overcome their fear of snakes. Overall, the students had a positive attitude towards first aid but lacked knowledge of snakebite emergencies. Public health awareness is required to dispel first-aid myths about snakebites and misconceptions regarding snakes.

## 1. Introduction

First aid is instant care given to the affected person before medical help arrives [1] The main goal of first aid is to prevent deterioration, promote recovery, and save lives. The rapidity with which first aid is provided is crucial for the survival of the individual [2]. Snakebites are rare in urban regions but rampant in rural areas of Saudi Arabia [3] and could occur when accidentally stepping on a snake while walking in the countryside, working in the fields, or playing on the grass. Snakes are of two types—venomous and non-venomous. Even though their appearance might seem the same to the victim, their outcome differs; one induces harm by injecting its venom into the body, while the other only frightens the victim. In both cases, first aid is a life-saving service [4].

More than 3500 species of snakes exist worldwide, of which around 17% are venomous, causing an annual death toll between 20,000 and 125,000. In addition, thousands develop chronic disabilities, such as amputation and blindness [5,6]. Snakebite incidence varies widely across geographical regions, but accurate estimates are difficult to obtain due to insufficient documentation by healthcare practitioners and health facilities, lack of collation of cases by central health authorities, and lack of formal documentation among traditional healers [1,5].

Human–snake interactions have always resulted in negative outcomes for humans and snakes alike. As far as agriculture workers are concerned, snakebites are emerging as an occupational hazard [7]. If anyone is bitten by a venomous or suspected venomous snake, it is vital to transport the affected person to the hospital as quickly as possible. The Hemotoxins in snake venom can have deleterious effects on the circulatory system, usually by attacking the body’s clotting capabilities. In cases of cytotoxic envenoming, there is usually painful swelling at the site, progressive blistering, and bruising, leading to localized tissue damage if left untreated [8,9,10]. First aid for snakebites delays the systemic absorption and the spread of the venom while transporting the victim to a nearby hospital for adequate treatment [11,12].

Despite their disrepute, snakes contribute to biodiversity and are an essential part of natural ecosystems [13]. The ecological value of snakes is now recognized, as they are crucial in reducing the number of rodents, which spread disease and destroy crops [14]. More importantly, the venom of a variety of snakes has medicinal qualities. The anti-tumor effects of some snake venoms [15] have been examined in addition to their anti-bacterial and pain-relieving properties. Hemotoxins have been used to treat heart attacks and blood diseases. The neurotoxins in snake venom are utilized to develop treatments for Alzheimer’s disease, Parkinson’s disease, stroke, and brain trauma [11,12,16,17]. As a result, not only do snakes play a vital role in nature, but they are also very important to humans due to the medical relevance of their venom. Apart from their use in producing anti-venom, snake venoms are a rich source of enzymes, peptides, and proteins with significant pharmacological activities [18]; hence, people’s attitudes toward them should change.

In Saudi Arabia, most snakebites are reported by agricultural and desert dwellers. These dwellers have limited access to medical care. In addition, students attending health-related courses at Jazan University come from distant parts of the province. They can be trained to be first-aid providers in areas where medical care is not easily accessible, as well as to educate and impart knowledge about the significance of snakes in nature. Students of health-related courses can change the health scenario of society if they are properly trained. However, they need to have basic skills and knowledge to minimize the injuries that are due to snakebites and save lives. Hence, the current study was undertaken to assess the knowledge and attitudes of upper-level students (3rd year and above) of health-related courses at Jazan University on performing first aid for snakebite emergencies and their perception of snakes.

## 2. Materials and Methods

The study was designed to assess the knowledge of handling snakebite emergencies and the attitudes while providing first aid, as well as students’ perceptions of snakes among medical, pharmacy, emergency medicine, and nursing students at Jazan University, which is a public university located in the Jazan province. Jazan province is one of the 13 provinces in Saudi Arabia, and it is located in the southwest region of the country [19].

### 2.1. Inclusion and Exclusion Criteria

To be included in this study, the participants had to be senior students (third year and above) in the medical, pharmacy, nursing, or emergency medicine of applied medicine colleges of Jazan University. Students who were in their first and second years of college or who returned incomplete questionnaires were not included in the study.

### 2.2. Study Design

This present study was descriptive, cross-sectional, and questionnaire-based, which employed convenience sampling to recruit participants from the health science colleges of Jazan University, Saudi Arabia. The participants included students studying in their third or higher year of the medical, pharmacy, nursing, and medical emergency programs of applied medicine at Jazan University. The study was conducted from January 2019 to March 2019.

### 2.3. Study Questionnaire

The items in the questionnaire were designed based on the management of snakebites, the general awareness of snakes in the surrounding area, and the general perception of snakes [20,21]. The questionnaire met all of the first-aid requirements outlined by the WHO protocol for treating snakebites [22]. After its preparation, the questionnaire was sent to three experts, one each from the College of Medicine, the College of Nursing, and the College of Emergency Medicine, to verify the content and face validity of the questionnaire. After that, the questionnaire was translated into Arabic by a professional expert. At the end, the Arabic version of the questionnaire was forward–backward translated to confirm its correctness and uniformity across the languages. The internal consistency of the different sections of the questionnaire was evaluated by the Cronbach’s alpha coefficient, where a value of 0.7 and above was considered satisfactory [23,24,25]. The Cronbach’s alpha values of 0.82, 0.87, and 0.84 for the knowledge, attitudes, and perception questions indicated excellent internal consistency. The pilot testing of the questionnaire was performed on a focus group of fifteen students, who were not included in the final analysis [26], and the suggestions received were discussed among all of the authors and incorporated into the final version of the questionnaire.

The questionnaire comprised 34 items, which were divided into two main sections. The first section included 5 questions pertaining to the participants’ demographic characteristics (gender, age, college, and year of study). The last question in this section was close-ended (“Yes”/”No”) and asked the respondents if they had received any training for snakebite management.

The second section of the questionnaire was further divided into 5 sub-sections. The first sub-section comprised 19 close-ended (“Yes”/”No”, “I Don’t Know”) questions related to the first-aid knowledge of snakebite emergencies, the identification of venomous and non-venomous snakes, and the immediate response to snakebites. The second sub-section tested the attitudes of the respondents with 6 close-ended (“Yes”/”No”, “I Don’t Know”) questions. Each item represented the individual’s attitude in a specific situation described by the statement. For the knowledge and attitudes sub-sections, the Bloom cut-off point [24,27] was employed to determine the level of knowledge or attitude, where >80% of correct responses were considered “Good”, 60% to 80% were considered “Moderate”, and <60% were deemed “Poor”. For each correct answer, the response was coded as 1, and an incorrect/I don’t know response was coded as 0. For the negative questions, a correct response (“No”) was coded as 1, and an incorrect response (“Yes” or “I Don’t Know”) was coded as 0.

The third sub-section included a single question on a 5-point Likert scale, which required the respondents to state the extent of their fear of snakes in the wild. The fourth and fifth sub-sections pertained to the perception of health science students’ perceptions towards encountering different types of snakes. We asked the respondents to indicate what action they would take if they encountered snakes in the wild and in their home or yard, respectively. The snakes were categorized as large venomous, small venomous, large non-venomous, and small non-venomous.

The last sub-section included a single open-ended question eliciting different methods or approaches the respondents believed would increase their support or respect for snakes. Based on the open-ended responses received, we grouped them under 7 categories.

### 2.4. Data Collection

For ease of data collection, the Arabic version of the questionnaire was entered onto Google Forms. Convenience sampling was used for the participant recruitment. The online questionnaire link was then sent as an email and through various student groups. The questionnaire link was accompanied by an invitation for all upper-level male and female students to participate in the study, who were also requested to forward it to their fellow students in their course.

The first page of the online version of the questionnaire informed the respondents of the background and objectives of the study. It was clearly mentioned that participation in the survey was completely voluntary and that the respondents were free to withdraw at any time they wanted. The respondents were also informed that all of their responses would remain confidential and anonymous and that the data generated from the study would only be used for scientific purposes. The inclusion criteria were also mentioned to ensure appropriate participation and they had to agree to the informed consent in order to proceed to the other sections of the questionnaire [26,28,29].

### 2.5. Statistical Analysis

The data from the Google Forms were first entered into Microsoft Excel and then exported to the Statistical Package for Social Sciences, Version 23.0 (SPSS 23.0) for further analysis. The participants’ demographics were analyzed using descriptive statistics, which were then expressed as frequencies and total percentages. The significant associations between the variables were assessed using the Pearson’s chi-square test. The level of significance was set at *p* < 0.05.

### 2.6. Ethical Considerations

The study was conducted according to the ethical considerations of the university, which was in line with the Declaration of Helsinki. This research protocol was reviewed and approved by the institutional ethics committee of Jazan University; approval number REC41/1-053. In addition, consent was obtained from the study subjects prior to answering the questionnaire.

## 3. Results

### 3.1. Internal Consistency Measures: Cronbach’s Alpha

The reliability of the different sections of the questionnaire was assessed using the Cronbach’s alpha coefficient, which measures the internal consistency; the results are shown in Table 1. The alpha coefficients for all of the sections of the questionnaire were found to be greater than 0.82 (i.e., above the 0.70 threshold), indicating an excellent degree of internal consistency and reliability of the developed questionnaire. The reliability studies were carried out on the pilot sample, which was subsequently deleted from the final analysis [24].

Table 2 summarizes the description of the sample, with more males (55%) than females responding to the questionnaire. Most of the respondents (86.4%) were within the age group between 20 and 25 years, followed by respondents over 25 years (8.1%), while those below 20 years were only 5.5%. Most of the respondents were undertaking the pharmacy program (44%), followed by nursing (25.2%), medicine (16.8%), and emergency medicine (13.9%). The sample represents a good distribution across the year of study, with 31.4% fourth-year students, 22.3% fifth-year students, 21% third-year students, 18.4% emergency medicine team members, and 6.8% sixth-year students. In addition, most of the respondents reported that they had received training in the management of snakebites (81.6%).

### 3.2. Knowledge about Snakebite First-Aid Management

The respondents’ knowledge regarding snakebite first-aid management is depicted in Table 3. The majority of the respondents could differentiate between different types of snakes (93.5%), thought that first aid is essential in the case of a snakebite (95.8%), and knew that the snakebite victim should be transported to the hospital as soon as possible (92.6%). Nearly two-thirds (65%) knew that snakebite victims should immobilize the bitten part. The least knowledge pertained to the question regarding “dry snakebite”, where only 8.4% of the respondents had knowledge regarding it.

Table 4 depicts the descriptive statistics of the knowledge categories, whereas Table 5 illustrates the association between the participants’ demographics and the different categories of the participants’ knowledge (good, moderate, and poor) on first-aid management in snakebite emergencies. The different categories of knowledge were found to be significantly associated with the gender of the participants (χ^2^ = 8.28; *p* = 0.02), their college (χ^2^ = 25.38; *p* < 0.001), and their year of study (χ^2^ = 12.07; *p* = 0.03). The participants’ age group was not significantly associated with their knowledge of first-aid management in snakebites.

### 3.3. Attitude towards Providing First-Aid Treatment to Snakebite Victims

As presented in Table 6, the majority of the respondents (95.15%) displayed a positive attitude towards attaining knowledge about the first-aid management of snakebites. However, less than two-thirds (59.87%) reported feeling tense while performing first aid in an emergency condition. Most of the respondents (85.76%) reported having respect for snakes. For the question “Have you ever killed a snake?”, nearly two-thirds (64%) of the participants responded negatively.

Table 7 shows the descriptive statistics of the three categories of attitude. Table 8 depicts the association between the participants’ demographics and the different categories of attitude of the participants (good, moderate, and poor) about first-aid management in snakebites. A significant association between the different categories of attitude was found for gender (χ^2^ = 25.76; *p* < 0.001) and college (χ^2^ = 16.10; *p* = 0.009). The year of study and age group were not significantly associated with the participants’ attitude to first-aid management in snakebites.

Figure 1 depicts the general perception of participants as how safe or afraid they would feel around a snake in the wild. As presented in Figure 2, the most likely perception among the students upon encountering a large venomous or large non-venomous snake in the wild is to ask someone to kill the snake (49.8% and 30.7%, respectively) or kill the snake themselves if it were a small venomous snake (28.2%). Around one-fourth of the respondents would let a large non-venomous snake go away (25.9%), while a lesser number would scare off the snake into the bush (20.4%) or try to capture the snake (8.4%). The most likely practical attitude among the students if encountered by a small non-venomous snake in the wild was to let the snake go away (31.7%) or ask someone to kill it (25.2%).

Figure 3 indicates the perceptions of the students if encountered by a large venomous snake in their yard or home. A little more than a quarter said that they would try to kill the snake themselves (26.9%) and around half would ask someone to kill it (51.5%). In the case of encountering a small venomous snake in their yard or home, their responses were also relatively similar, wherein they would ask someone to kill the snake (43.7%) or try to kill the snake themselves (35%).

As shown in Figure 4, the most prominent approaches that would increase respect for snakes among the students were learning to handle snakes safely (35%), followed by seminars on snake behavior, anatomy, and benefits of snakes (32.7%).

## 4. Discussion

The present cross-sectional, questionnaire-based study was conducted among the students at the health science colleges of Jazan University with the aim of assessing their knowledge and attitudes toward providing first-aid services to snakebite victims and their perception towards snakes in general. The essence of providing first aid during snakebites is to conserve life, prevent further injury, and speed up recovery.

On the basis of the responses provided by the participants, for some key questions assessing their knowledge, it was shown that a good percentage gave the correct answer. For example, when asked “Do you advise the snakebite victim to immobilize the bitten part?”, nearly two-thirds of the participants (65%) responded positively, which was the correct response. The same applies to the question “Do you advise massaging the bitten part?”, for which the correct response was given by close to two-thirds (62.8%) of the participants; thus, demonstrating good knowledge of first aid for snakebite victims and its emergency management. This study, however, has also revealed a gap in the students’ knowledge of first-aid treatments for snakebites, as indicated by their responses to questions pertaining to advising tourniquets, sucking venom out of the wound, and knowing what a dry bite is, which could help determine the level of training that should be provided in the future, evaluating myths and misconceptions about snakes and snakebites among students, and promoting respect for snakes in the ecosystem.

More participation was seen from the students in the pharmacy program than from the students of medicine, emergency medicine, and nursing. A higher proportion of the students (81.6%) had received training in providing first aid for snakebites during their study program. As a first step in preparing the students for community service, Jazan University, Saudi Arabia, has adopted and incorporated essential life-saving skills into the graduate curriculum. In contrast to many universities located in urban areas and other parts of the Gulf, the students of Jazan University have the advantage of receiving first-aid training for snakebite emergencies [30,31].

The majority of the students (more than 90%) were aware of the basics of first aid, such as its importance, the different types of snakes, and transportation of the victim to the hospital, as indicated by their responses to these questions. This could be due to the training received on first aid for snakebites by these students. Just over half (54%) of the study participants said “Yes” to the question of tying a tourniquet near the snakebite area. These findings are similar to those of Subedi et al., who reported a very high percentage of their sample saying “Yes” to the application of a tourniquet near the site of the snakebite [32]. In rural areas, people generally apply a tourniquet unscientifically to reduce the blood flow at the bitten site, which might result in devastating outcomes [1]. In contrast, more than half of the medical students in India think that it is good to use a tourniquet as a first-aid strategy for snakebite victims [33]. A study found that more than half of the snakebite victims used ineffective and harmful first aid in South Asia, with tourniquets being used by a larger proportion of the patients [34]. When used around the proximal sections of a limb, tight (arterial) tourniquets can cause significant discomfort due to the progressive development of ischemia on the limbs, which can lead to gangrene if left in place for a long time [22]. More than half (62%) of the health science college students at Jazan University, Saudi Arabia, were unsure about massaging the bite location. Only a meagre 6.1% of the participants responded correctly to the question “Do you advise massaging the bitten part?”, and this was the question with the lowest correct responses. This highlights a key knowledge gap in the first-aid management of snakebites. Massage at the bitten place improves blood circulation, and the injected venom would be absorbed faster; hence, it is not advisable to massage the bitten area or increase the mobility of the affected part. Similar results were observed among Nepalese medical students, who believed various myths and misconceptions were part of effective snakebite treatments [32]. Medical students in Nepal were aware of applying pressure immobilization or bandages before enrolling in medical programs [35].

Similarly, the participants provided a skeptical response to questions pertaining to incision at the bitten site for blood oozing, sucking the venom by a healthy volunteer, applying an electric shock to the bitten region, and applying herbal medicines. Incising, rubbing, vigorously washing, using chemicals and herbs, and other methods of interfering with the snakebite wound can result in infection, improve absorption of the snake venom, and increase local bleeding. Even though traditional remedies are outdated for snakebites, many students were still confused about the idea, which is evident by their responses, where less than two-thirds (61.8%) responded with an “I Don’t Know” response to this question. Regarding the question about sucking out the venom by a healthy volunteer, less than half (43.4%) gave the correct answer. The students probably believe that sucking or slitting the affected area will stop the venom from spreading throughout the body. Similar findings were seen in other research studies, where students were more likely to remove poison from the body during the first-aid procedure [36,37,38,39,40].

Furthermore, most of the students were unsure about dry snakebites. This is another question where students gave the least correct answers (8.4%). Although dry bites are still delivered by venomous snakes, they are different as the venom is not injected into the body (as opposed to a bite by a non-venomous snake). Unfortunately, most of the time, the victims’ conduct will be similar as they are unaware that they have been bitten by a non-venomous snake. The resemblance in bite response could be the source of confusion. Fear, not venom, causes the reactions to dry bites. The students will benefit from more clarity on venomous and dry bites, which will aid them in saving many lives that perish due to fear. Hence, knowing the different types of snakes present in the locality would help to provide appropriate first-aid management.

The students of the emergency medicine program had a much higher level of knowledge compared to the other health science college students, plausibly due to their hospital exposure to emergencies. It was shown that more than three-fourths (85.7%) of the fifth- and sixth-year study participants displayed good knowledge, possibly due to their professional maturity.

More than half of the students (59.9%) responded in affirmative that they would feel tense while performing first aid to a snakebite victim in an emergency situation. This seems understandable, as the majority of the students (92.9%) did not feel good about snakes in general. Feeling uncomfortable at the thought of snakes might lead to a stressful situation during a snakebite scenario. This was corroborated by the responses of the participants about their perception of encountering different types of snakes in the wild, home, or yard. Nearly half of them stated that they would ask someone to kill large or small venomous snakes if they encountered them in their home or yard (51.5% and 43.7%, respectively). When asked what they would do if they came across a snake in the wild, they gave similar unfavorable replies. Only a small percentage of the students reported that they would let a non-venomous snake go if they came across it in the wild, at home, or in their yards. Snakes are one of nature’s wondrous creatures, and they do not attack unless their habitats are disturbed, or they are accidentally stepped on. Most of the time, they would do it as a form of defense. Snakes are vital to people for various reasons, including keeping rodents at bay and offering therapeutic value; thus, they must be conserved. According to estimations, around 17% of more than 3500 snake species identified worldwide are venomous [6], with the rest being non-venomous. Antivenom therapy is the ultimate treatment when a snakebite is linked with envenomation and should be delivered appropriately (Regional Office for South-East Asia 2016). People have a proclivity to kill snakes, which is especially true when the snake bites [40].

Snakes contribute to the ecological balance and should be protected. Yet, in our survey, 48.5% of the respondents said that snakes should be killed when they bite, 21.4% disagreed, and 30.1% were undecided. This suggests that the students lack the knowledge of snake conservation in the ecosystem and their medical significance.

Learning through seminars about the safe handling of snakes and playing video games with snake protagonists are among the different activities suggested to enhance respect for snakes. However, the most important and positive finding was that they all exhibited a strong desire to learn snakebite first-aid management.

## 5. Strength and Limitations

This was the first study in the Jazan province to test the expertise of medical, pharmacy, nursing, and emergency medical students at Jazan University on providing first-aid treatments to snakebite patients and managing emergencies. If properly trained, these students could be valuable in preserving the lives of residents of rural areas while also instructing them on how to conserve snakes to preserve biodiversity.

One of the study’s limitations is that we only tested the participants’ academic knowledge of snakebites without examining their practical skills, particularly in terms of first aid. Additionally, as our study population belonged to a single university in a single province, it makes it difficult to generalize the findings to other parts of the country. Due to the convenience sampling, the study participants might not have been representative. Moreover, we did not measure the confidence level of the students in dealing with snakes. Furthermore, a self-reported survey has a potential for recall bias, and convenience sampling increases the potential for selection bias. 

## 6. Conclusions

In conclusion, our research demonstrated gaps in the knowledge on how to deal with snakebites appropriately and scientifically along with a lack of awareness about how vital this subject is for medical, pharmacy, nursing, and emergency medicine students. This research also revealed that the students did not fully comprehend the role of snakes in the ecosystem and their medical impact. Given the significant chance that health science students may have to deal with snakebites in their professional or personal lives, we recommend the inclusion of snakebite training and management in the curriculum of the health science programs. Our findings also identified areas where knowledge gaps existed for our university students, stressing the necessity for future research into the understanding and practices of health science students in other universities in Saudi Arabia, along with the knowledge and awareness of healthcare professionals on the first-aid management of snakebites.

## Figures and Tables

**Figure 1 healthcare-10-02226-f001:**
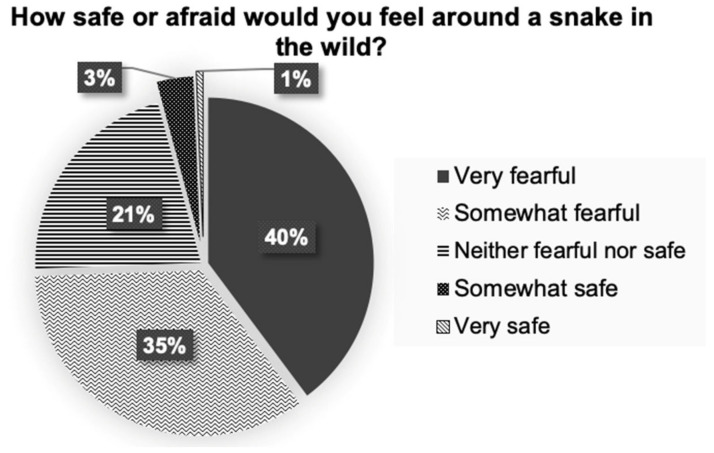
Participants’ perceptions of snakes in the wild.

**Figure 2 healthcare-10-02226-f002:**
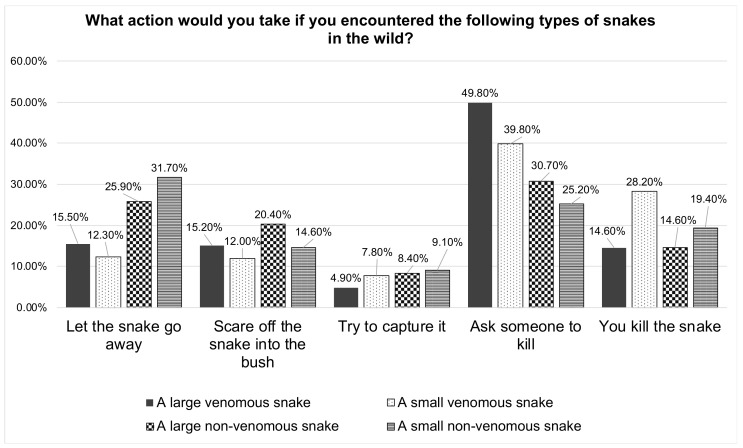
Participants’ perceptions towards encountering different types of snakes in the wild.

**Figure 3 healthcare-10-02226-f003:**
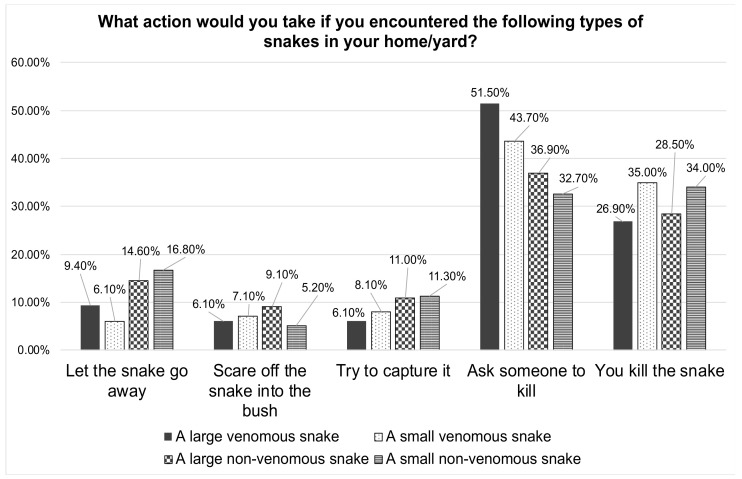
Participants’ perceptions towards encountering different types of snakes in their home or yard.

**Figure 4 healthcare-10-02226-f004:**
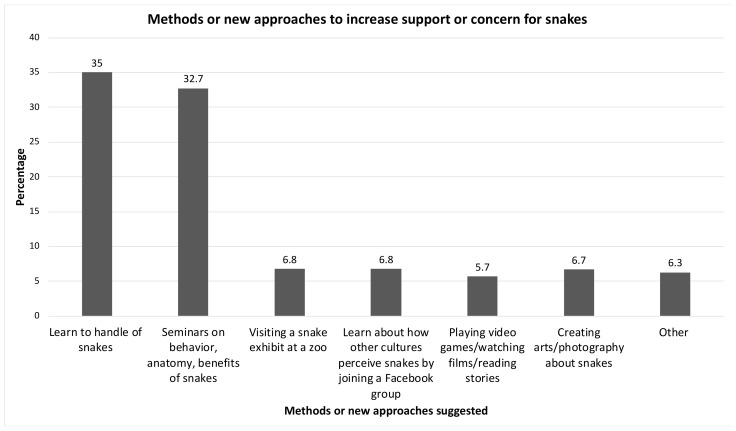
Methods and new approaches that would increase respect for snakes.

**Table 1 healthcare-10-02226-t001:** Reliability Analysis.

Section	Cronbach’s Alpha
Knowledge about snakebite first-aid management	0.83
Attitude about first-aid management and snakes	0.87
Perception towards encountering types of snakes in the wild, home/yard	0.84

**Table 2 healthcare-10-02226-t002:** Respondents’ Characteristics.

Demographic Details	Variable	*N* (%)
Gender	Male	170 (55)
	Female	139 (45)
Age	<20 years	17 (5.5)
	20–25 years	267 (86.4)
	>25 years	25 (8.1)
Profession	Medical	52 (16.8)
	Nursing	78 (25.2)
	Pharmacist	136 (44)
	Emergency medicine	43 (13.9)
Year of study	Third year	65 (21)
	Fourth year	97 (31.4)
	Fifth year	69 (22.3)
	Sixth year/Internship	78 (25.2)
Training received in the management of snakebite	Yes	252 (81.6)
	No	57 (18.4)

**Table 3 healthcare-10-02226-t003:** Participants’ knowledge about snakebite first-aid management.

Items	Yes%	No%	I Don’t Know%
Do you think “first aid” is essential in snakebite cases?	95.80	1.30	2.90
Do you know that there are two types of snakes–a) venomous and non-venomous?	93.50	3.20	3.20
Should the snakebite patient be transported to the hospital soon after the bite?	92.60	1.30	6.10
Do you know the ambulance number to be dialed during an emergency?	71.20	20.40	8.40
Have you ever heard the word “first aid” in the management of snakebite?	67.30	22.70	10.00
Do you advise the snakebite victim to immobilize the bitten part?	65.00	7.80	27.20
Do you know the differences between venomous and non-venomous snakebites?	42.40	46.00	11.70
Should pressure immobilization bandages be applied around the bite site?	40.80	27.80	31.40
Do you think that intramuscular injection of analgesics can be performed for first aid in snakebite victims?	32.70	19.10	48.20
Do you know who is most bitten by venomous snakes?	17.20	68.60	14.20
Do you know of “dry snakebite”?	8.40	79.60	12.00
* Do you advise massaging the bitten part?	6.10	62.80	31.10
* Should healthy volunteers suck the venom out of the wound?	24.60	43.40	32.00
* Do you advise to suck the bitten site soon after snakebite to remove poison?	37.50	36.60	25.90
* Do you advise to incise or pricks/punctures the bite site soon after the snakebite?	27.20	35.00	37.90
* Is electric shock at the site of snakebite useful	3.90	30.40	65.70
* Is it required to kill the snake after it bites the victim?	48.50	21.40	30.10
* Do you advise tying tight bands (tourniquet method) soon after the snakebite?	54.00	20.70	25.20
* Is topical instillation or application of herbs beneficial?	23.60	14.60	61.80

* indicates questions with negative connotations and were reverse scored (No = 1, Yes = 0, I don’t know = 0).

**Table 4 healthcare-10-02226-t004:** Descriptive statistics of the knowledge categories.

Knowledge Category	*N* (%)
Good Knowledge	7 (2.3)
Moderate Knowledge	90 (29.1)
Poor Knowledge	212 (68.6)

**Table 5 healthcare-10-02226-t005:** Cross-tabulations between the participants’ demographics and the knowledge categories.

Variables	Poor Knowledge	Moderate Knowledge	Good Knowledge	Total *N* (%)	(Chi Square) *χ*^2^	*p*-Value
Gender						
Male	105 (49.5%)	60 (66.7%)	5 (71.4%)	170 (55.0%)	8.28	*p* = 0.02
Female	107 (50.5%)	30 (33.3%)	2 (28.6%)	139 (45.0%)		
College						
Medical	44 (20.8%)	6 (6.7%)	2 (28.6%)	52 (16.8%)	25.38	*p* < 0.001
Nursing	60 (28.3%)	17 (18.9%)	1 (14.3%)	78 (25.2%)		
Pharmacy	88 (41.5%)	47 (52.2%)	1 (14.3%)	136 (44.0%)		
Emergency Medicine	20 (9.4%)	20 (22.2%)	3 (42.9%)	43 (13.9%)		
Year of Study						
Third Year	51 (24.1%)	13 (14.4%)	1 (14.3%)	65 (21.0%)	12.07	*p* = 0.03
Fourth Year	72 (34.0%)	25 (27.8%)	0 (0.0%)	97 (31.4%)		
Fifth Year	42 (19.8%)	25(27.8%)	2 (28.6%)	69 (22.3%)		
Sixth Year	47 (22.2%)	27 (30.0%)	4 (57.1%)	78 (25.2%)		
Age Group						
<20	14 (6.6%)	3 (3.3%)	0 (0.0%)	17 (5.5%)	1.74	*p* = 0.74
20–25	182 (85.8%)	78 (86.7%)	7 (100.0%)	267 (86.4%)		
>25	16 (7.5%)	9 (10.0%)	0 (0.0%)	25 (8.1%)		

**Table 6 healthcare-10-02226-t006:** Participants’ attitude towards snakebite first-aid management.

	Yes%	No%	I Don’t Know%
Do you feel tense while performing first aid in an emergency condition?	59.87	27.51	12.62
Have you ever touched a snake?	31.72	66.99	1.29
Do you feel good about snakes in general?	2.91	92.88	4.21
Do you have respect for snakes?	85.76	11.33	2.91
Have you ever killed a snake?	34.30	64.08	1.62
Will you show interest in attaining knowledge about first aid?	95.15	2.27	2.59

**Table 7 healthcare-10-02226-t007:** Descriptive statistics of the attitude categories.

Attitude Category	*N* (%)
Good Attitude	16 (5.2)
Moderate Attitude	280 (80.9)
Poor Attitude	43 (13.9)

**Table 8 healthcare-10-02226-t008:** Cross-tabulations between the participants’ demographics and the attitude categories.

Variables	Poor Attitude	Moderate Attitude	Good Attitude	Total *N* (%)	(Chi Square) *χ*^2^	*p*-Value
Gender						
Male	38 (88.4%)	126 (50.4%)	6 (37.5%)	170 (55.0%)	25.76	*p* < 0.001
Female	5 (11.6%)	124 (49.6%)	10 (62.5%)	139 (45.0%)		
College						
Medical	9 (20.9%)	40 (16.0%)	30 (18.8%)	52 (16.8%)	16.10	*p* = 0.009
Nursing	4 (9.3%)	68 (27.2%)	6 (37.5%)	78 (25.2%)		
Pharmacy	17 (39.5%)	114 (45.6%)	5 (31.3%)	136 (44.0%)		
Emergency Medicine	13 (30.2%)	28 (11.2%)	2 (12.5%)	43 (13.9%)		
Year of Study						
Third Year	8 (18.6%)	55 (22.0%)	2 (12.5%)	65 (21.0%)	9.61	*p* = 0.13
Fourth Year	17 (39.5%)	72 (28.8%)	8 (50.0%)	97 (31.4%)		
Fifth Year	5 (11.6%)	63 (25.2%)	1 (6.3%)	69 (22.3%)		
Sixth Year	13 (30.2%)	60 (24.0%)	5 (31.3%)	78 (25.2%)		
Age Group						
<20	2 (4.7%)	14 (5.6%)	1 (6.3%)	17 (5.5%)	0.95	*p* = 0.95
20–25	39 (90.7%)	214 (85.6%)	14 (87.5%)	26 (86.4%)		
>25	2 (4.7%)	22 (8.8%)	1 (6.3%)	25 (8.1%)		

## Data Availability

Data of the present study can be made applicable upon reasonable request to the corresponding author (S.S.A.).

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
