# Peer review of "Knowledge and Attitude of First-Aid Treatments for Snakebites, and the Perception of Snakes among the Students of Health Sciences at Jazan University, Saudi Arabia"

_healthcare, 2022, doi:10.3390/healthcare10112226_

Round 1

Author Response

We would like to thank the editor and the reviewers for the constructive review of our manuscript entitled “Knowledge and attitude of first aid treatment for snake bites, and the perception of snakes among the students of health sciences of Jazan University, Saudi Arabia.”. The comments have significantly improved the quality of the manuscript and the authors are greatly thankful for the opportunity to re-submit the manuscript.

We have attempted to address all the issues raised by the learned editor and reviewers in our revised

manuscript. Also, the manuscript has been proofread for the English language by a native English speaker.

We are submitting the revised manuscript with all changes highlighted in yellow color. Point-by-point responses are tabulated below, and the line numbers mentioned in the responses are referred to  the revised manuscript. We hope that the revised version of the manuscript will be considered for publication in your esteemed journal.

Reviewer 2 Report

Dear Author

Thank you for submitting your manuscript  to Healthcare. This is a very well written paper with detailed discussion and appropriate conclusion. The topic is significant.  My comments are below:

1. In table 5, 1st question "Don't you feel tense while performing first aid in an emergency condition?". All other questions start with do you....? Why does this question start with "don't you". This would create confusion in the answer.

2. Lines 206 and 207, check and rewrite again. "...or kill the snake by themselves 206 (28.2%). A smaller percentage will let the snake go away (25.9%)". You started explaining about large venomous snake and large non-venomous snake, but 28.2% is for small venomous snake. 25.9% is for large non-venomous snake and not for large venomous snake. Explain clearly.

3. Line 215 - you mentioned "Results in Table 7 indicate the perception of the health professional students if they encountered a large venomous snake in their yard or home". Actually this was depicted in table 6 and not table 7.

4. Line 216 - A little more than a quarter said they would ask someone to kill the snake or try to kill the snake themselves (26.9%). Actually in the table, ask someone to kill the snake was 51.5%.

5. Remove table 7, since the same details were depicted in figure 2.

Author Response

We would like to thank the editor and the reviewers for the constructive review of our manuscript entitled “Knowledge and attitude of first aid treatment for snake bites, and the perception of snakes among the students of health sciences of Jazan University, Saudi Arabia.”. The comments have significantly improved the quality of the manuscript and the authors are greatly thankful for the opportunity to re- submit the manuscript.

We have attempted to address all the issues raised by the learned editor and reviewers in our revised

manuscript. Also, the manuscript has been proofread for English language by a native English speaker.

We are submitting the revised manuscript with all changes highlighted in yellow color. Point-by- point responses are tabulated below, and the line numbers mentioned in the responses are referred to the revised manuscript. We hope that the revised version of the manuscript will be considered for publication in your esteemed journal.

Round 2

Reviewer 1 Report

This is an outstanding revision.  The manuscript’s writing and presentation are substantially improved – the language flows smoothly, the explanation of the goals, methods, and results are clear and consistent.  The new scoring system for snakebite attitude and knowledge is very clear and logical, all analyses are now explained well.  The new tables are easy to read and interpret.  Overall this reads well and is easy to follow throughout, it is a strong paper that clearly presents useful information.

I have just a few suggestions, presented sequentially below.  Most are editorial in nature and very minor changes or requests for clarification.  Others are suggestions for how the authors MIGHT consider revising their presentation of some results (e.g. figures instead of tables), but these truly are suggestions and not requirements.  The authors may have their reasons for preferring a table format, and that’s fine.

I congratulate the authors on their excellent work.

131 – Change to “venomous and nonvenomous”

147 – Change “listed out” to “categorized” or “classified”

181 – I still think a brief explanation of Cronbach’s alpha is warranted here.  It can be as simple as “…using Cronbach’s alpha, which measures…”

Table 3 – This is really useful information.  You might consider reorganizing the questions in order of the % reporting correctly.  That will help visually convey the story of where the knowledge and lack of knowledge is.  Using the order originally presented on the questionnaire doesn’t seem critical.

Table 4, Table 6 – These tables are extremely useful and very well organized.  I do think that presenting these data in graphical form (perhaps a series of pie charts like Figure 1) would be more effective, as graphs are generally more efficient than tables at conveying percentage information quickly and intuitively.  If you elect to stick with a table format, these are quite good.  In either case, I recommend you start by adding a section showing knowledge/attitude of students overall, then break that down by gender, year, etc. as already you do here.

214 – I would be a little careful with the explanation here.  Technically your Chi-square analysis just shows a statistical association between snakebite knowledge and sex.  The fact that males showed a greater knowledge is a post hoc explanation of WHY you got that result, but it is not shown by the statistical result per se.  The same with field of study and year of study.  It’s a subtle but important (in my opinion) difference.

227 – Couldn’t the lack of having killed a snake also mean they just never encountered one?  Or that they dislike them so much the had to have someone else kill it for them?

232-233 – See my previous comments about interpreting Chi-square tests of association.

Figure 1, Figure 2 – Excellent.  These communicate your results simply, effectively, and immediately intuitively.  For Fig. 1, you might want to consider using a gray-scale color format for those who will print out your manuscript.  That’s probably also dependent on journal standards.

Table 7 – I would move the “what action would you take in the wild” line to UNDER the “Let snake go/scare off/etc” headings for easier reading.  Otherwise my suggestion is much like that for Table 4 and 6… this information might be better presented in two grouped bar graphs, but this table itself is very good.

272-279 – This is a really useful summary.  In my opinion it emphasizes the usefulness of including overall results for knowledge and attitude earlier (e.g. in Tables 4 and 6) and reorganizing the question order in Table 3.

285-287 – I don’t understand.  Researchers as in the authors?  And they were in the “same program” meaning Pharmacy?  Why would that affect your response rate?

335-336 – I would point out that “dry” bites are still delivered by venomous snakes, just that no venom is actually injected (as opposed to a bite by a snake that is not venomous at all).

340 – Change to “venomous”

Author Response

We would like to sincerely thank the Reviewer 2 for the positive and encouraging comments after the first revision of our manuscript entitled “Knowledge and attitude of first aid treatment for snake bites, and the perception of snakes among the students of health sciences of Jazan University, Saudi Arabia.”. We are greatly thankful to the editor for the opportunity to re-submit the manuscript after the second round of minor revisions.

We have attempted to address all the issues raised by the learned editor and reviewers in our revised manuscript. We are submitting the revised manuscript with all new changes highlighted in blue color. Point-by-point responses are tabulated below, and the line numbers mentioned in the responses are referred to in the revised manuscript. We hope that the revised version of the manuscript will be considered for publication in your esteemed journal.

Kindly check the attached file.
